

# Response of hydrology and CO₂ flux to experimentally-altered rainfall frequency in a temperate poor fen, southern Ontario, Canada

Danielle D. Radu[1], Tim P. Duval[1]

[1]Department of Geography, University of Toronto Mississauga, Mississauga, ON, Canada, L5L 1C6

*Correspondence to*: Tim P. Duval (tim.duval@utoronto.ca)

**Abstract.** Predicted changes to the precipitation regime in many parts of the world include intensifying the distribution into lower frequency, large magnitude events. The corresponding alterations to the soil moisture regime may affect plant growth and soil respiration, particularly in peatlands, where large stores of organic carbon are due to gross ecosystem productivity (GEP) exceeding ecosystem respiration (ER). This study uses a combined lab and field approach to examine the effect of

changing rainfall frequency on peatland moisture controls on $CO_2$ uptake in an undisturbed cool temperate poor fen. Lab monoliths and field plots containing mosses, sedges, or shrubs received either 2.3, 1, or 0.5 events per week, with total rainfall held constant. Decreasing rain frequency led to lower near-surface volumetric moisture content (VMC), water table (WT), and soil tension for all vegetation types, with minimal effect on evapotranspiration. The presence of sedges in particular led to soil tensions > -100 cm of water of a sizeable duration (37 %) of the experiment. Altered rainfall frequencies affected GEP but had

little effect on ER: overall low-frequency rain led to reduced net $CO_2$ uptake for all three vegetation types. VMC had a strong control on GEP and net ecosystem exchange (NEE) of the *Sphagnum capillifolium* monoliths, and decreasing rainfall frequency influenced these relationships. Overall, communities dominated by mosses became net sources of $CO_2$ after three days without rain, whereas sedge communities remained net sinks for up to 14 days without rain. Results of this study demonstrate the hydrological controls of peatland $CO_2$ exchange dynamics influenced by changing precipitation frequency

and suggest these predicted changes in frequency will lead to increased vascular plant growth and limit the carbon-sink function of peatlands.

## 1 Introduction

Northern peatlands cover less than 3% of the Earth's land surface but have sequestered approximately one-third of the global store of terrestrial carbon (Gorham, 1991; Yu, 2012), serving an important role in the global carbon cycle (Roulet et al., 2007;

Limpens et al., 2008). However, this accumulated carbon is expected to be severely affected by climate change due to changes in temperature and precipitation patterns (Kettles and Tarnocai, 1999; Li et al., 2007; Gallego-Sala and Prentice, 2013). The net uptake of carbon in peatlands is due to relatively cool and moist conditions that promote slower release of $CO_2$ by ecosystem respiration (ER) than uptake by gross ecosystem productivity (GEP) (Frolking et al., 2002; Rydin and Jeglum, 2006; Frolking et al., 2010). Maintenance of the carbon sink function of peatlands is dependent on a high water table (WT) to minimize



respiration (Alm et al., 1999; Strack et al., 2006; Riutta et al., 2007) and high near-surface soil moisture to enable high rates of *Sphagnum* moss photosynthesis (Waddington and Roulet, 2000; Moore et al., 2002; Adkinson and Humphreys, 2011), both of which are controlled by the precipitation regime. Changing climate and weather patterns are leading to an intensification of the precipitation regime (Easterling et al., 2000; Cao and Ma, 2009; Diffenbaugh and Field, 2013), with increasingly-longer periods without rain that threaten the sink function of northern peatlands (Nijp et al., 2017).

Climate projections for boreal and temperate regions of the northern hemisphere suggest larger but less frequent precipitation events, with limited to no increases in absolute seasonal and annual precipitation (Trenberth, 2011; IPCC, 2013; Sillmann et al., 2013; Wang et al., 2014; Westra et al., 2014). This repackaging of the precipitation regime, especially during the growing seasons, is expected to lead to decreases in near-surface soil moisture, increased soil moisture variability, and deeper WT (Knapp et al., 2002; Gerten  et al., 2008; Vervoort and van der Zee, 2008; Wu et al., 2012). Lower WT and near-surface soil

moisture due to seasonal drought or temperature-induced increased ET have been shown to lead to reduced net carbon uptake and/or greater rates of $CO_2$ emission from peatlands (Carroll and Crill, 1997; Alm et al., 1999; Tuittila et al., 2004; Strack et al., 2006; Riutta et al., 2007). Additionally, changes to peatland hydrology through drought have led to changes in vegetation biomass and community composition, which in turn can alter peatland $CO_2$ exchange depending on the photosynthesis to respiration ratio of the newly established communities (Buttler et al, 2015; Potvin et al., 2015; Churchill et al., 2015; Dieleman

et al, 2015). However, these experimental and monitoring studies typically reduce total precipitation input during the study period, while much less research has focussed on redistribution of rainfall whilst maintaining precipitation totals.

Most studies examining the impact of precipitation frequency on ecosystems have focussed on grassland systems (Hoover et al., 2014; Knapp et al., 2015; Wilcox et al., 2015; Didiano et al., 2016), with comparatively fewer studies in wetland environments. Riparian marsh species' biomass accumulation is negatively affected by month-long periods without added

precipitation (Garssen et al., 2014). In peatland systems, shifts in rainfall frequency have been shown to affect net primary production from *Sphagnum* moss (Robroek et al., 2009; Nijp et al., 2014) and methane emission (Radu and Duval, 2017). *Sphagnum* moss photosynthesis in particular responds quickly to trace amounts of precipitation input (Strack and Price, 2009; Adkinson and Humphries, 2011); however, precipitation inputs are only available to these peat-building species for 2-3 days before it is evaporated (Ketcheson and Price, 2014). In the absence of precipitation, *Sphagnum* depends on capillary rise from

the saturated zone to the photosynthesizing capitula (Clymo and Hayward, 1982).  When WTs are too low and precipitation is absent, soil water tensions increase and hyaline cells drain, causing desiccation and reduced photosynthesis (Thompson and Waddington, 2008; Strack et al., 2009; McCarter and Price, 2014).

In addition to the non-vascular *Sphagna* species, peatlands can be dominated by shrub and graminoid plant functional types, primarily ericaceous shrubs and sedges, respectively (Rydin and Jeglum, 2006). Vascular plants photosynthesize as long as

the component cells retain turgor pressure, and are less susceptible to periods of low precipitation and WT drawdowns (Malmer et al., 1994; Vile et al., 2011). Peatland shrubs typically have increased productivity when WTs are lowered (Weltzin et al.,



2001, Murphy et al., 2009, Bragazza et al., 2013; Munir et al., 2015) due to their shallow rooting system (Wallén, 1986). Sedges have deep roots (40-100 cm below peat surface) aided by aerenchyma that transport oxygen to lower depths (Silvan et al., 2004) and are thus tolerant of high WT conditions, but have also been documented to perform well under low WT conditions (Fenner et al., 2007; Dieleman et al., 2015). Additionally, plant and soil respiration rates differ between moss, sedge, and shrub communities (Chimner, 2004; Juszczak et al., 2012; Duval and Radu, 2017). Since vascular plants can comprise a significant portion of peatland productivity and respiration (Szumigalski and Bayley, 1996; Moore et al., 2002; Riutta et al. 2007; Korrensalo et al., 2017), their responses to climate change in concert with *Sphagnum* must be taken into account when assessing peatland carbon cycling.

Previous studies of the effect of lowered water table on different plant functional types to study the impacts of climate change on peatland carbon cycling have not included the importance of precipitation frequency (Riutta et al., 2007; Churchill et al., 2015; Potvin et al., 2015). Recent research on the interaction between peatland hydrology, carbon cycling, and precipitation frequency has focussed on *Sphagnum* moss in lab and/or modelling studies (Nijp et al., 2014; Nijp et al., 2017). There exists a research gap at the intersection of precipitation frequency and its effect on peatland hydrology and carbon cycling for a variety of peatland communities. Therefore, the objectives of this study are to i) investigate the effect of changing precipitation frequency on peatland hydrology including WT position, volumetric moisture content, soil water tension, and evapotranspiration in three common peatland vegetation communities – *Sphagnum* moss only, Sedge with *Sphagnum*, and *Sphagnum* with ericaceous shrubs; and ii) determine the relationship between hydrologic conditions under the different rainfall frequency treatments and GEP, ER, net ecosystem $CO_2$ exchange of those communities. We altered precipitation frequency without changing total precipitation amount in both *in situ* field plots and lab monoliths through commensurate changes to event magnitude.

## 2 Methods

### 2.1 Study Site

The study was carried out in an undisturbed poor fen in southern Ontario, Canada (44°15'13.34" N, 80°20'46.83" W). Vegetation was dominated by *Sphagnum* moss (particularly *S. capillifolium* but also *S. rubellum*, *S. fuscum*, and *S. magellanicum*), sedges (*Carex oligosperma* and *Eriophorum vaginatum*) and shrubs (mostly *Chamaedaphne calyculata*, as well as *Rhododendron groenlandicum* and *Vaccinium uliginosum*). Moss ground cover is near 100 % throughout the site, except in areas with mature shrubs, where ground cover averaged 15 %. Average peat depth in the sample area is 2.1 m overlaying a sandy silt till substrate (Burwasser, 1974). The climate near the site is characterised by a mean annual temperature of 6.4 °C and a mean annual precipitation of 996 mm (1981-2010 normal at Ruskview, ON station, data available:





http://climate.weather.gc.ca/climate.normals/). Rainfall events > 0.2 mm occurs on 43 % of the days during the early May – end of September growing season at this climate station.

## 2.2 Field Experiment

We examined the effect of rain frequency on peatland hydrology and $CO_2$ exchange among different vegetation communities

in the field with the use of rainout shelters. The rainout shelters were built to 3 m X 3 m the previous summer to our study distributed among the three most common occurring plant communities in the peatland: 1) *Sphagnum* moss (mainly *S. capillifolium*) ("Moss" plots) 2) *Sphagnum* with sedges (mainly *Carex oligosperma*) ("Sedge" plots), and 3) ericaceous shrubs (mainly *Chamaedaphne calyculata*) with minimal *Sphagnum* ground cover ("Shrub" plots). We covered the shelters with new 6-mil transparent polyethylene sheeting immediately prior to our field campaign to exclude and collect natural rainwater to be

used for our treatments. The 9 plots within each community were assigned one of three rain frequency treatments, replicated three times: 3 events/week, 1 event/week, and 1 event/2 weeks, hereafter referred to as "High-Frequency" (HiFreq), "Medium-Frequency" (MedFreq), and "Low-Frequency" (LowFreq), respectively. Although the frequency of the rain events was different, the total amount of water in each 2-week period was equal between treatments. Further details on the site setup can be found in Radu and Duval (2017) and Radu and Duval (submitted).

$CO_2$ fluxes were determined using the dynamic closed chamber method (Alm et al., 1999). Net ecosystem exchange (NEE) was measured using a 60 cm X 60 cm X 30 cm clear Plexiglass chamber with attached fan placed over permanent, ridge aluminum collars sealed with water, and an EGM-4 infrared gas analyzer (IRGA) (PP Systems, Massachusetts, USA). Photosythetically-active radiation (PAR) and air temperature in the chamber were measured with a QSO-S PAR Photon Flux sensor (Apogee Instruments, Utah, USA) and the IRGA, respectively. Plots were sampled at least weekly throughout the study

period (at midday (1000-1700 h) in both full sunlight and cloudy conditions throughout the season. Ecosystem respiration (ER) was measured using the same technique using an opaque aluminum chamber. Gross ecosystem productivity (GEP) was calculated by subtracting ER from NEE. We used the convention that negative numbers denote ecosystem $CO_2$ uptake and positive numbers denote ecosystem $CO_2$ loss to the atmosphere.

Perforated PVC wells covered with 250-µm Nitex mesh were inserted into the ground to a depth of 1 m within each sample

plot to monitor the WT level. EC-5 sensors (Decagon Devices Inc., Washington, USA) were carefully inserted vertically into the peat for a composite depth of 0-5 cm within sample plots receiving each rain treatment in each of the vegetation communities (VMC).

## 2.3 Lab Experiment

Intact peat cores (30 cm diameter X 40 cm height) were collected from the peatland to investigate how precipitation frequency

affects $CO_2$ exchange under controlled climate and WT regimes. Details of core collection and setup can be found in Radu and



Duval (2017). Briefly, cores of each of the three vegetation communities were placed in an environment-controlled chamber (FXC-19 Chamber, BioChambers, Winnipeg, Manitoba, Canada) and water tables were kept at -5 cm for an acclimatization period to the chamber conditions. Climate conditions of the chamber can be found in Table S1.

We manipulated rainfall frequency over a four-month study period. Three precipitation frequency treatments were randomly assigned to monoliths of each vegetation community type: 3 events/week ('HiFreq-Lab'), 1 event/week ('MedFreq-Lab'), and 1 event/2 weeks ('LowFreq-Lab'). Simulated rainwater was a diluted Rudolph nutrient solution (Rudolph et al., 1988; Faubert and Rochefort, 2002) to limit detrimental effects to *Sphagnum* moss growth (Dieleman et al., 2015). The amount of water added for the individual events was adjusted such that the total amount of water was the same between treatments for each two-week period (see Table S2 for treatment details). At the beginning of the study period, all water levels were set to -5 cm ('High') and after two months were adjusted to -15 cm ('Low'). Within each of these two 2-month periods, WT positions were allowed to naturally fluctuate with the addition of rainwater and the loss of water through ET to simulate field conditions.

Perforated PVC wells (1.27 cm diameter) covered with 250-µm Nitex mesh were carefully inserted into peat cores to monitor WT levels; measurements were made three times weekly in all monoliths. VMC was measured with EC-5 soil moisture sensors installed vertically into the peat of each monolith for a composite depth of 0-5 cm; data were measured at half-hourly intervals with EM50 data loggers. Soil water tension was measured using 15-cm elbow tensiometers (Soil Measurement Systems, Arizona, USA) installed horizontally at 5 cm below moss surface; an SMS INFIELD 7 portable tensicorder was used for to measure tension 3 times per week. Actual evapotranspiration (ET) was measured directly by weighing the monoliths before and after rainwater solution additions throughout the study period.

$CO_2$ exchange was measured three times per week in the environmental chamber as above, though under constant atmospheric conditions (Table S1). $CO_2$ fluxes were measured using a clear Plexiglas cylindrical chamber (30 cm diameter X 40 cm height). A fan was mounted on the inside of the chamber to mix the air and homogenize the $CO_2$ concentration during measurements. NEE was measured by fitting the clear chamber to a monolith and sealed with petroleum jelly to ensure an air-tight seal. The chamber was connected to an EGM-4 IRGA. Ecosystem respiration (ER) was measured with an opaque aluminum cover placed over the chamber to exclude PAR. Gross ecosystem productivity (GEP) was calculated by subtracting ER from NEE.

**2.4 Data Analysis**

The data were checked for homogeneity and normality with Levene's test and Shapiro-Wilk's test, respectively. Differences in hydrological and $CO_2$ exchange parameters between rainfall frequency treatments and vegetation communities were assessed with ANOVA. Post-hoc pairwise comparisons of the mean responses to different treatments were assessed with Tukey tests. Relationships between hydrological variables and $CO_2$ exchange components were examined with linear and nonlinear regression. All statistical analyses were performed using the Statistica 8 software package (StatSoft Inc.).



## 3 Results

### 3.1 Peatland hydrology under changing rainfall frequency

Our rainout shelters in the field were able to significantly change the incident precipitation between our three rainfall treatments, with the HiFreq treatment being similar to the ambient rainfall regime during the study periods (Table S2). These

changes to incident rain affected the near-surface volumetric soil moisture but had limited effect on WT fluctuation. Seasonal VMC was significantly higher in the HiFreq treatment than the LowFreq treatment for the Moss and Shrub vegetation communities ($p < 0.001$; Table 1). Conversely, average VMC was lowest in the HiFreq treatments of the Sedge plots ($p < 0.01$). Moreover, VMC in the field was more variable as rainfall frequency decreased, with ranges of ~24, 27, and 32 % for HiFreq, MedFreq, and LowFreq plots, respectively. On the other hand, WT fluctuation under the rainout shelters was heavily

influenced by WT levels in the surrounding peatland. Shrub WT did not differ between rainfall treatments, while the HiFreq treatment WT was significantly lower than the two lower-frequency treatments in both the Moss ($p < 0.05$) and Sedge ($p < 0.001$) communities (Table 1), which mimicked the pattern found outside the shelters in the areas selected for the HiFreq treatments. Thus, our efforts to restrict lateral flow from outside the shelters to our study areas had limited success.

In the lab peat monolith experiment, decreasing precipitation frequency generally resulted in decreased WT depth, near-surface

VMC, and soil tension for all vegetation types, regardless of initial WT position (Table 2). Water table fluctuation between rainfall treatments for the different vegetation monoliths are shown in Figure S1. During the high WT period of the experiment, WT depths were significantly lower in the LowFreq-Lab relative to the HighFreq-Lab treatment in all vegetation communities ($p < 0.001$; Table 2). In the second phase of the experiment when WTs were reset to -15 cm the WT was significantly lower in the LowFreq-Lab relative to the HiFreq-Lab treatments in the Sedge + Moss ($p < 0.01$) and Moss ($p < 0.05$) monoliths, but

not different in the Moss + Shrub communities.

Near-surface VMC followed the same pattern of WT fluctuation between rainfall treatments throughout the experiment for all vegetation monoliths (Fig. S2). For all vegetation types VMC was > 60 % in the HiFreq-Lab monoliths for more than half the experiment (Fig. 1). In comparison, the 50th quantile of VMC in the LowFreq-Lab treatment was much lower at 49, 51, and 52 % for the Moss, Sedge + Moss, and Moss + Shrub, respectively (Fig. 1). Overall, VMC was significantly higher in the HiFreq-

Lab than the LowFreq-Lab treatment for the Moss and Moss + Shrub monoliths for both portions of the experiment ($p<0.001$; Table 2). In the Sedge + Moss monoliths there were no significant differences in VMC between treatments under the high WT period ($p = 0.298$), but during the low WT period VMC in the LowFreq-Lab treatment was significantly lower than either of the more-frequent treatments ($p < 0.005$). There was a significant relationship between VMC and WT for all vegetation communities in the lab study ($R^2 = 0.77\text{-}0.93$, $p < 0.0001$; Fig. 2). Additionally, there were differences in the proportional

decrease in VMC for a unit decrease in WT between rain treatments for all vegetation types. VMC decreased by 0.1-0.2 % more in the LowFreq-Lab relative to the HighFreq-Lab treatment per cm drop in the WT in all vegetation communities.



Increasing the duration since the last precipitation event led to significant decreases in VMC for all vegetation communities (Fig. 3). The number of consecutive dry days had a greater effect on VMC declines during the high-WT phase of the experiment for both the Moss and Moss + Shrub monoliths. The VMC rate of decline decreased in the Moss monoliths by 50 % between

the two phases of the experiment, from 1.6 % d$^{-1}$ during the high-WT period ($R^2 = 0.40$; $p < 0.001$) to 0. 8% d$^{-1}$ ($R^2 = 0.19$; $p < 0.001$) during the low-WT phase (Fig. 3a). The decrease in rate of VMC decline was ~31 % between high- and low-WT portions of the experiment for the Moss + Shrub monoliths (Fig. 3c). In contrast the rate of VMC decline with increasing consecutive dry days increased slightly in the Sedge + Moss monoliths as the WT was lowered, from 1.6 % d$^{-1}$ during the high phase ($R^2 = 0.25$; $p < 0.001$) to 1.8% d$^{-1}$ during the low-WT phase ($R^2 = 0.27$; $p < 0.001$; Fig. 3b).

Average near-surface soil tension increased (became more negative) with decreasing rain frequency in all vegetation treatments (Table 2). These tensions rarely reached -100 cm, the critical level for *Sphagnum* capillary water supply, except in the Sedge + Moss monoliths subject to the LowFreq-Lab treatment during the low-WT period, where soil tension frequently was -150 cm. The monoliths experienced linear increases in tension of ~ -1.1, -1.8 and -1.4 cm d$^{-1}$ of no rainfall under high WT levels for Mosses ($R^2 = 0.28$; $p < 0.001$), Sedge + Moss ($R^2 = 0.22$; $p < 0.001$), and Moss + Sedge ($R^2 = 0.36$; $p < 0.001$), respectively

(Fig.4). These rates of tension increase generally remained the same in the low-WT portion of the experiment for the Moss and Moss + Shrub monoliths (Fig. 4a,c). On the other hand, soil tension increased at a rate of -4.3 cm d$^{-1}$ without rain in the Sedge + Moss monoliths during the low-WT phase, more than double the rate under high WT (Fig. 4b).

Sedge + Moss evapotranspiration (ET) was significantly higher under HiFreq-Lab and LowFreq-Lab than MedFreq-Lab treatments during both the high- and low-WT phases of the lab experiment ($p < 0.05$; Table 2). There were no significant

differences in ET in the Moss and Moss + Shrub monoliths. There was no clear trend between ET and the number of days since rainfall; however, ET exceeded 3.5 mm d$^{-1}$ for up to two days after rainfall in all vegetation communities, and generally remained below 3 mm d$^{-1}$ for periods up to 14 days without rain.

### 3.2 CO$_2$ exchange dynamics

Figure 5 demonstrates the rainfall treatments affected CO$_2$ exchange in all three vegetation communities. Gross ecosystem

productivity from the Moss monoliths decreased with decreasing rainfall frequency under low-WT, with rates nearly twice as high in the HiFreq-Lab treatment (-0.101 ±0.010 mg CO$_2$ m$^2$ s$^{-1}$) compared to the LowFreq-Lab treatment (-0.056 ±0.014 mg CO$_2$ m$^2$ s$^{-1}$; $p = 0.017$; Fig. 5a). While there were no differences in Moss ER between rainfall treatments during high WT the LowFreq-Lab treatment led to significantly more respiration during the low-WT period than either higher-frequency treatments ($p = 0.004$). Moss NEE significantly decreased (less CO$_2$ uptake/more CO$_2$ release) with decreasing precipitation frequency

in both WT treatments. Under the high-WT period, Moss monoliths were net CO$_2$ sinks, with NEE 3.5-times lower in the LowFreq-Lab (-0.010 ±0.008 mg CO$_2$ m$^2$ s$^{-1}$) relative to the HiFreq-Lab treatment (-0.038 ±0.006 mg CO$_2$ m$^2$ s$^{-1}$; $p = 0.013$). During the low WT treatment, the Mosses switched to net CO$_2$ sources, with the LowFreq-Lab treatment emitting between



three- and four-times more $CO_2$ (0.123 ±0.016 mg $CO_2$ $m^2$ $s^{-1}$) than the Med- (0.040 ±0.007 mg $CO_2$ $m^2$ $s^{-1}$) and HiFreq-Lab (0.031 ±0.004 mg $CO_2$ $m^2$ $s^{-1}$) treatments (p < 0.001; Fig. 5a).

The presence of shrubs with the moss increased GEP in the LowFreq-Lab treatment during the low-WT phase, such that there were no differences in GEP due to rainfall treatment at this time (Fig. 5b). Additionally, during high WT the MedFreq-Lab treatment for the Moss + Shrub monoliths led to more $CO_2$ uptake than the LowFreq-Lab treatment. There were no differences in ER between vegetation treatments for either portion of the experiment for the Moss + Shrub monoliths. The pattern of NEE in response to rainfall frequency for Moss + Shrub monoliths was similar that observed with Moss – under high WT all three
treatments were $CO_2$ sinks, with a shift to sources under low WT (Fig. 5b). While there were no differences in rates of NEE between frequency treatments under high WT (p = 0.412), under low WT the MedFreq-Lab (0.042 ±0.007 mg $CO_2$ $m^2$ $s^{-1}$) and LowFreq-Lab (0.07 ±0.011 mg $CO_2$ $m^2$ $s^{-1}$) were nearly six- and 10-times greater sources of $CO_2$ than the HiFreq-Lab treatment (0.007 ±0.008 mg $CO_2$ $m^2$ $s^{-1}$; p = 0.019).

There was a trend of greater $CO_2$ uptake with decreasing rainfall frequency under high WT in the Sedge + Moss monoliths,
with LowFreq-Lab resulting in greater GEP (-0.155 ±0.008 mg $CO_2$ $m^2$ $s^{-1}$) than HiFreq-Lab (-0.098 ±0.002 mg $CO_2$ $m^2$ $s^{-1}$; p = 0.03; Fig. 5c). Under low WT, GEP more than doubled in the HiFreq-Lab treatment to -0.214 ±0.012 mg $CO_2$ $m^2$ $s^{-1}$, which was significantly higher than the MedFreq-Lab treatment (-0.146 ±0.007 mg $CO_2$ $m^2$ $s^{-1}$; p = 0.002). While there were no significant differences in ER for the two WT levels, the relative rank order of the three rainfall frequency treatments remained the same as for GEP. Overall, under high WT there was no effect of rain frequency on NEE for the Sedge + Moss monoliths;
however, under low WT there was significantly greater NEE during frequent rain (-0.045 ±0.012 mg $CO_2$ $m^2$ $s^{-1}$; p = 0.002) as compared to MedFreq-Lab (0.008 ±0.007 mg $CO_2$ $m^2$ $s^{-1}$) and LowFreq-Lab (-0.009 ±0.012 mg $CO_2$ $m^2$ $s^{-1}$), both of which were not statistically different from zero (Fig. 5c).

### 3.3 Controls on $CO_2$ exchange between rainfall frequency treatments

There was a clear difference in the ratio of GEP between LowFreq and HiFreq treatments across the range of PAR values
measured in the field (Fig. 6). The Low:High Frequency GEP ratio was < 1.0 for over two-thirds of the Moss GEP measurements, and was only > 1.5 when PAR was < 800 μmol $m^2$ $s^{-1}$. In contrast, GEP in the LowFreq treatment exceeded the HiFreq measurements at equal PAR for 75 % of the Shrub measurements, with ratios > 2.0 a common occurrence under high light conditions (PAR > 800 μmol $m^2$ $s^{-1}$). There was no trend to the ratio of GEP between low- and high-frequency treatments in the Sedge communities, with values between 0.5 and 1.5 across all light levels (Fig. 6).

There were significant unimodal relationships between Moss GEP and VMC from the monoliths, and these relationships varied between the rain frequency treatments (Fig. 7a). The relationship was strongest for the high frequency rain treatments, with peak rates of GEP occurring at 65 % VMC. The strength of the relationship decreased with decreasing rain frequency, as did the VMC at which peak GEP occurred. There were no significant relationships between GEP and WT. The position of the WT



was highly correlated with Moss ER for all of the rain frequency treatments ($p < 0.001$; Fig. 7b). As rain frequency decreased,

WT declines led to proportionally greater rates of ER, with the rate of increase in LowFreq-Lab ER being 1.5-times higher than the HiFreq-Lab treatment (Fig. 7b). Moss NEE was strongly controlled by near-surface VMC, with significant quadratic correlations for the HiFreq-Lab and MedFreq-Lab treatments, and a third-order polynomial relationship for the LowFreq-Lab rainfall treatment (Fig. 7c). The switch from net $CO_2$ uptake to a net source occurred in the Moss monoliths at VMC of 60, 62, and 64 % in the HiFreq-Lab, MedFreq-Lab, and LowFreq-Lab, respectively. VMC was below these levels for 49 % of the

study in the HiFreq-Lab treatment, 66 % in the MedFreq-Lab, and 73 % in the LowFreq-Lab treatment (Fig. 1). Similar relationships between moss carbon flux and peatland hydrology between rain frequency treatments were found in the field experiment (Fig. S3); however, positive NEE occurred at higher VMC, which occurred for much less of the study period (20 % in the LowFreq treatment).

Under the HiFreq-Lab treatment VMC had a unimodal control on rates of GEP from the Sedge + Moss monoliths, with peak

gross production occurring at a VMC of 49 % (Fig. 8a). Decreasing rain frequency rendered this relationship insignificant. Ecosystem respiration was influenced by WT position in the Sedge + Moss monoliths, with rates increasing between 57 and 71 % faster with WT drawdown in the LowFreq-Lab and MedFreq-Lab, respectively, than for the HiFreq-Lab conditions (Fig. 8b). There were moderate but highly significant ($p < 0.001$) linear correlations between WT and Sedge + Moss NEE for the lower-frequency treatments; a quadratic relationship existed between NEE and WT in the HiFreq-Lab treatment, with

minimum rates of net $CO_2$ uptake occurring when the WT was at -29 cm (Fig. 8c). The trends with the Moss + Shrub monoliths were similar to the Moss-only monoliths (data not shown). There were no significant relationships between WT or VMC with GEP or NEE from the Shrub communities in the field, though WT did control ER in these communities ($p < 0.001$).

Overall, increasing the number of days since the last rainfall event led to a strong increase in NEE ($CO_2$ efflux to the atmosphere) for the Moss monoliths ($R^2 = 0.31$, $p < 0.01$), a moderate increase for the Moss + Shrub monliths ($R^2 = 0.18$, $p

<0.001$), but almost no increase in the Sedge + Moss monoliths ($R^2 = 0.05$, $p = 0.044$; Fig. 9). This relationship was linear for the vascular plant communities, and quadratic for the moss-only monoliths. The Moss and Moss + Shrub monoliths had a net uptake of $CO_2$ when the duration since the last event was under 3.5 days; however, after this threshold, there was net emission of $CO_2$. On the other hand, Sedge + Moss monoliths were largely sinks of $CO_2$ for up to 2 weeks without rainfall, and were predicted to have a net emission of $CO_2$ after 15 days between events.

**4 Discussion**

**4.1 Plant community-mediated response of peat hydrology to precipitation frequency**

It is expected that a shift in the precipitation to larger, more infrequent events will lead to lower WT levels and drier surface conditions (Knapp et al., 2008; Piao et al., 2009). We found that decreasing precipitation frequency while holding total seasonal





rain constant resulted in lower average WT positions, lower VMC, and higher soil tension in all vegetation communities
(Tables 1, 2; Fig. S1, S2). The smaller, more frequent precipitation events were able to buffer against seasonal WT declines
and maintain more moisture in the near-surface peat layer (Fig. 2). On the other hand, the larger, less frequent events
contributed moisture to deeper layers, which led to the observed increased rates of VMC decline with concomitant WT decline
(Fig. 2). Therefore, in addition to climate change leading to increased ET and lower WT positions in peatlands (Whittington
and Price, 2006; Munir et al., 2015), the frequency of rainfall events is likely to contribute to even drier surface conditions
than is currently considered.

Decreasing rainfall frequency allowed for continued near-surface soil moisture decreases (Fig. 3) and tension
increases (Fig. 4). Delivering rain every three days, the average for the site (Table S2), prevented these large soil moisture
declines in communities dominated by Sphagnum moss (Table 1; Fig. S2). On the other hand, the presence of sedges increased
ET relative to the moss-only communities. Vascular plant abundance increases peatland ET, even during periods of limited
rainfall (Takagi et al., 1999; Petrone et al., 2004; Admiral and Lafleur, 2007; Wu et al., 2013; Takashi et al., 2016). During the
first portion of the experiment when WTs were high, the rate of drying was the same between Moss and Sedge + Moss
monoliths; however, during the low-WT portion the rate of drying in the sedge community became 75 % higher than in the
moss-only community (Fig. 3a, b). This led to over a four-fold greater rate of tension increase in the presence of sedges (Fig.
4a, b). The low-frequency rain treatment in the Sedge + Moss community led to near-surface tensions < -100 cm, sometimes
after only four days without rain. This tension threshold is generally considered the point at which Sphagnum can no longer
effectively photosynthesize (Price, 1997, Thompson and Waddington, 2008).

In our study -100 cm of tension was reached at a VMC of 37 %, similar to thresholds found in other peatlands (Price
and Whitehead, 2004; Cagampan and Waddington, 2008). The high- and medium-frequency treatments for all vegetation
communities maintained soil moisture above this value for at least 92 % of the experiment; however, deviations below this
threshold increased considerably in the low-frequency rain treatment (Fig. 1). The Moss and Moss + Shrub monoliths subject
to Low-Frequency rain experienced moss-level VMC < 37 % for ~10 % of the experiment, whereas the Sedge + Moss
monoliths were below this threshold for 38 % of the experiment. Overall, our results suggest that the interaction between
deeper WT and less-frequent precipitation regimes expected with climate change will lead to high tensions that may limit
moisture uptake by *Sphagnum* mosses in peatlands dominated by sedges.

**4.2 Precipitation frequency – peatland hydrology interactive effects on CO$_2$ exchange**

The precipitation regimes we imposed affected the CO$_2$ exchange of the vegetation communities. Overall, decreasing rainfall
frequency led to decreased NEE (less storage / greater flux to the atmosphere) from the peatland communities tested, with
differences in NEE for four of the six vegetation-WT combinations we tested (Fig. 5). Nijp et al. (2014) also found NEE
decreased from three *Sphagnum* species as precipitation frequency decreased. Field measurements of CO$_2$ fluxes from





peatlands have also documented decreased GEP and increased ER during extended rainless periods, switching peatlands to net sources of $CO_2$ during these short periods (Alm et al., 1999; Lund et al., 2012). Seasonal estimates from our study site confirm increased flux of $CO_2$ to the atmosphere from moss communities during low-frequency rain; however, the abundant shrub communities we were able to test in the field showed higher seasonal NEE (less source to the atmosphere) under low-frequency rain, due to higher GEP from the mature shrubs (Radu and Duval, submitted).

Lower WTs characteristic of drought periods decrease productivity and increase respiration, resulting in overall net loss of carbon to the atmosphere from *Sphagnum*-dominated peatlands (Carroll and Crill, 1997; Alm et al., 1999; Tuittila et al., 2004; Chivers et al., 2009, Potvin et al., 2015). Our study demonstrates low WTs exacerbate the effect of low rain frequency through greater rates of ER, causing the Moss and Moss + Shrub communities to switch from net sinks of $CO_2$ to net sources (Fig. 5a, b). Additionally, low WT resulted in the Sedge + Moss monoliths subject to the two lower frequency treatments to switch from
sinks to becoming carbon neutral (Fig. 5c). The lack of NEE response to low WT from the Sedge + Moss community subject to frequent rain was due to increased GEP, presumably because the high frequency rain maintained moss photosynthesis, while the sedges increased production with lower WT (Wu et al., 2013; Potvin et al., 2015).

Carbon assimilation by *S. capillifolium* was sensitive to rainfall frequency-induced changes in VMC. As rainfall frequency decreased, peak GEP rates were lower and occurred at lower VMC, while the strength of the relationship between GEP and
VMC weakened (Fig. 7). Repeated drying and wetting of mosses has been found to result in lower photosynthetic capacity due to damage to moss cellular integrity, including degradation of chlorophyll and rupture of cell membranes (Schipperges and Rydin, 1998). The low-frequency rain events allowed for greater near-surface VMC variability, which may have contributed to lower moss GEP at any given VMC. Gerdol et al. (1996) found *Sphagnum* species, including the dominant species in our study, *S. capillifolium*, were unable to resume photosynthesis after an 11-day period without new water. In our
study, the mosses continued to photosynthesize, albeit at lower rates, after 13 days without rain, as soil water tensions were weak enough (> -100 cm) to allow water uptake. Mosses subject to a regime of six days between rainfall events had GEP rates 25% lower than under a regime of two days between events.

The observed increase in the VMC threshold at which the *Sphagnum capillifolium* monoliths switched from NEE sinks to sources as precipitation frequency decreased (Fig. 7a) was related to the concomitant lower WTs (Fig. 2a). Therefore,
decreasing rain frequency not only lowered near-surface soil moisture but also created an additional stress for the mosses by reducing the availability of capillary water due to the lower WT. During the high-WT portion of the experiment VMC was found to drop below these thresholds after 10-11 days without rain; however, the low-WT period was characterized by near-surface VMC almost always less than 60 % (Fig. 3a). Strack et al. (2009) found a sink-source threshold for *S. rubellum* of ~45 % VMC in the near-surface peat, and Nijp et al. (2014) found a VMC of 48 % corresponded to a switch from sink to source
for *S. balticum*. All three species are lawn- or small hummock-forming species capable of withstanding low water availability, yet differ greatly in their response to VMC. The physiological mechanisms driving this range in moisture content threshold to



NEE among *Sphagna* species were beyond the scope of our study, but these differences in species-specific responses to soil water content are very important for parameterizing peatland ecosystem models (le Roux et al., 2013; Nijp et al., 2017).

In our study, moss-dominated communities became sources of $CO_2$ to the atmosphere after less than a week without rain (Fig. 9). This switch from carbon sink to source was primarily driven by decreased VMC between events limiting GEP (Fig. S2, 7a). The monoliths dominated by vascular plants were less affected by rainfall-induced decreases in VMC due to their deeper roots (Silvan et a., 2004), with the sedge communities maintaining uptake or carbon neutrality for the maximum 14 consecutive dry days of our experiment (Fig. 9). In our study region, southern Ontario, seven consecutive dry days occur quite regularly during the growing season, with one or two 14-day dry periods typical, and these dry periods are predicted to increase in both length and recurrence during growing seasons (Orlowsky and Seneviratine, 2012; Sillmann et al., 2013; Walsh et al., 2014).

### 4.3 Implications for peatland plant productivity and climate change

Our study demonstrates that vascular plants in peatlands will likely have a competitive advantage over mosses as rainfall becomes less frequent, particularly if accompanied by lower WT as a result of climate change. *Sphagnum*-only communities experienced a significant decrease in GEP with less frequent rainfall, while GEP in the communities with vascular plants was generally unaffected. Additionally, decreased precipitation frequency had stronger negative implications for *S. capillifolium* in the presence of sedges, with near-surface tensions > -100 cm after only 3 days without rainfall when the WT was deep (>-15 cm). Although we did not measure *S. capillifolium* productivity separate from the Sedges in the same communities, white, desiccated *S. capillifolium* capitula were observed in Sedge + Moss monoliths receiving less frequent rainfall when the water table was low. In a companion study we found low frequency rain led to significantly more seasonal sedge and ericaceous shrub cover and GEP than high frequency rain (Radu and Duval, submitted). Ericaceous shrubs in particular seem to thrive, with rates of GEP under LowFreq exceeding HiFreq rates 75 % of the time, generally at multiples of 2-7 under high light conditions (Fig. 6).

Our results show that decreased precipitation frequency will decrease net $CO_2$ uptake in peatland plant communities dominated by *Sphagnum*, both in the presence and absence of sedges and juvenile shrubs. With the lower water table positions expected as a result of increased ET, we showed that these communities may switch to $CO_2$ sources, and decreased precipitation frequency may increase net $CO_2$ release. Furthermore, increased shrub dominance (Hedwall et al., 2017) is likely to shift the $CO_2$ balance towards increased $CO_2$ efflux to the atmosphere. Therefore, we found that decreasing precipitation frequency is likely to lead to a positive feedback for climate change due to increased net $CO_2$ release to the atmosphere caused by drier surface conditions and a shift to sedge- and shrub-dominated communities.

### 5 Conclusions



Our study has demonstrated that decreasing rainfall frequency results in lower average VMC, WT, and soil water tension, and increases their variability. In turn, these changes in peat hydrology led to changes in $CO_2$ exchange dynamics, with significant effects on GEP and NEE. Moss-dominated communities in particular were strongly affected by changes in rainfall frequency, with decreasing gross and net $CO_2$ uptake with deceasing rain frequency. In contrast, sedge-dominated communities were able

to better withstand the longer periods without rain, and actually increased GEP with low-frequency rain when the WT was close to the surface. The empirical relationships between $CO_2$ exchange and near-surface VMC and WT showed clear differences between rainfall treatments, demonstrating the influence of rain frequency on these couplings. The presence of a low WT was found to have very strong effects on $CO_2$ exchange, shifting the Moss and Moss + Shrub communities from net $CO_2$ sinks to net sources while rendering the Sedge + Moss communities $CO_2$ neutral under the lower-frequency treatments.

We show the maintenance of near-surface moisture is very important to moss productivity, and as little as four consecutive dry days (CDD) was sufficient to illicit a net $CO_2$ efflux response from the moss monoliths. On the other hand, the deeper rooting system of the sedges prevented these monoliths from becoming sources for the extent of our experiment (14 CDD). Most climate change projections predict increases in the number and recurrence of CDD during the growing season for many peat-forming areas throughout the globe and our results suggest this will limit net $CO_2$ uptake in peatlands. We show the

presence of a lower WT, as predicted due to rising temperatures, increases the effect of lowered precipitation frequency. These rainfall-induced moisture and $CO_2$ dynamics should be included in peatland ecosystem and climate models.

**Code Availability**

N/A

**Data Availability**

Data are available from the corresponding author upon request

**Author contribution**

DDR and TPD developed the idea and methodology for the study, as well as led the experimental setup. DDR led the data acquisition. TPD and DDR performed the data analysis. Both authors wrote the manuscript.

**Competing Interests**

The authors declare that they do not have any conflicts of interest.



**Acknowledgements**

We thank the entire Stream and Wetland Ecohydrology Research Group at UTM for their assistance with the field portion of this study. In particular, we thank Ana Banjavcic and Michael Harris for help with core collection and William Ng for lab analysis. This research was funded by NSERC Discovery Grant #418197 to TPD and a NSERC CGS‑M and UTM-Graduate
Expansion Fund to DDR.

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




**Table 1:** Mean (standard deviation) values of hydrological variables for each precipitation treatment within each vegetation community during the field experiment (May-September 2015). Different letters indicate significant differences (p < 0.05) between precipitation treatments within each water table and vegetation treatment. No letters indicate no significant differences.

| Vegetation Community | Rainfall Treatment | VMC (%) | WT (cm) |
|---|---|---|---|
| Moss | HiFreq | 75(6)a | -17(6)a |
| | MedFreq | 75(7)a | -12(6)b |
| | LowFreq | 71(8)b | -13(6)ab |
| Shrub | HiFreq | 50(3)a | -18(6) |
| | MedFreq | 26(3)b | -18(6) |
| | LowFreq | 30(4)c | -16(6) |
| Sedge | HiFreq | 27(2)a | -21(6)a |
| | MedFreq | 82(6)b | -11(6)b |
| | LowFreq | 68(11)c | -13(6)b |



**Table 2:** Mean (standard deviation) values of hydrological variables for each precipitation, water table, and vegetation community treatment during the laboratory experiment. Different letters indicate significant differences (p < 0.05) between 625 precipitation treatments within each water table and vegetation treatment. No letters indicate no significant differences.

| Rainfall Treatment | VMC (%) | WT (cm) | ψ (cm) | ET (mm/d) | VMC (%) | WT (cm) | ψ (cm) | ET (mm/d) |
|---|---|---|---|---|---|---|---|---|
| | | High Water Table | | | | Low Water Table | | |
| | | | | Moss | | | | |
| HiFreq-Lab | 74(8)a | -6(1)a | -18(8)ab | 1.4(.07) | 50(8)a | -20(4)a | -25(5)a | 1.5(0.2) |
| MedFreq-Lab | 71(6)ab | -7(2)a | -16(7)a | 1.3(.07) | 47(6)ab | -20(4)a | -28(6)a | 1.3(.07) |
| LowFreq-Lab | 65(8)b | -11(3)b | -23(9)b | 1.5(0.1) | 43(7)b | -23(5)b | -32(9)b | 1.4(.08) |
| | | | | Moss + Shrub | | | | |
| HiFreq-Lab | 77(8)a | -8(2)a | -14(7)a | 1.4(0.2) | 50(7)a | -21(4) | -30(12) | 1.6(0.2) |
| MedFreq-Lab | 72(5)ab | -10(3)ab | -18(10)ab | 1.5(.09) | 46(8)ab | -22(4) | -28(10) | 1.4(0.1) |
| LowFreq-Lab | 68(5)b | -12(3)b | -21(10)b | 1.6(0.1) | 43(8)b | -22(5) | -33(14) | 1.5(0.1) |
| | | | | Sedge + Moss | | | | |
| HiFreq-Lab | 67(8) | -13(4)a | -21(10)a | 1.8(0.1)a | 44(9)a | -31(8)a | -41(11)a | 1.9(0.2) |
| MedFreq-Lab | 69(10) | -10(3)a | -22(9)a | 1.4(0.1)b | 48(6)a | -22(5)b | -29(5)a | 1.3(0.3) |
| LowFreq-Lab | 64(10) | -19(10)b | -33(15)b | 2.0(0.1)a | 37(10)b | -37(6)c | -71(38)b | 1.8(0.2) |








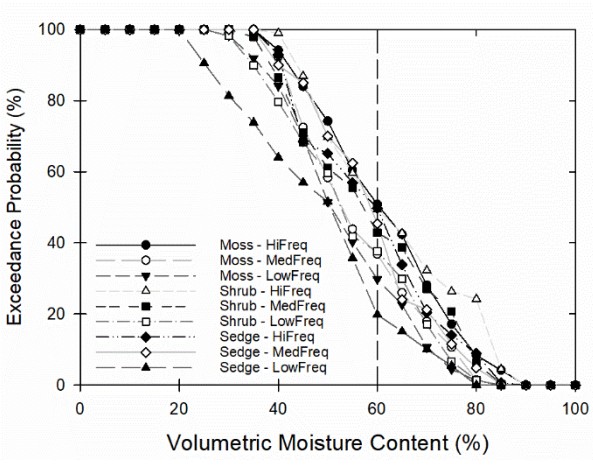

**Figure 1: Exceedance probability of VMC from the peat monoliths subject to Hi-, Med-, and LowFreq-Lab precipitation frequency treatments.**



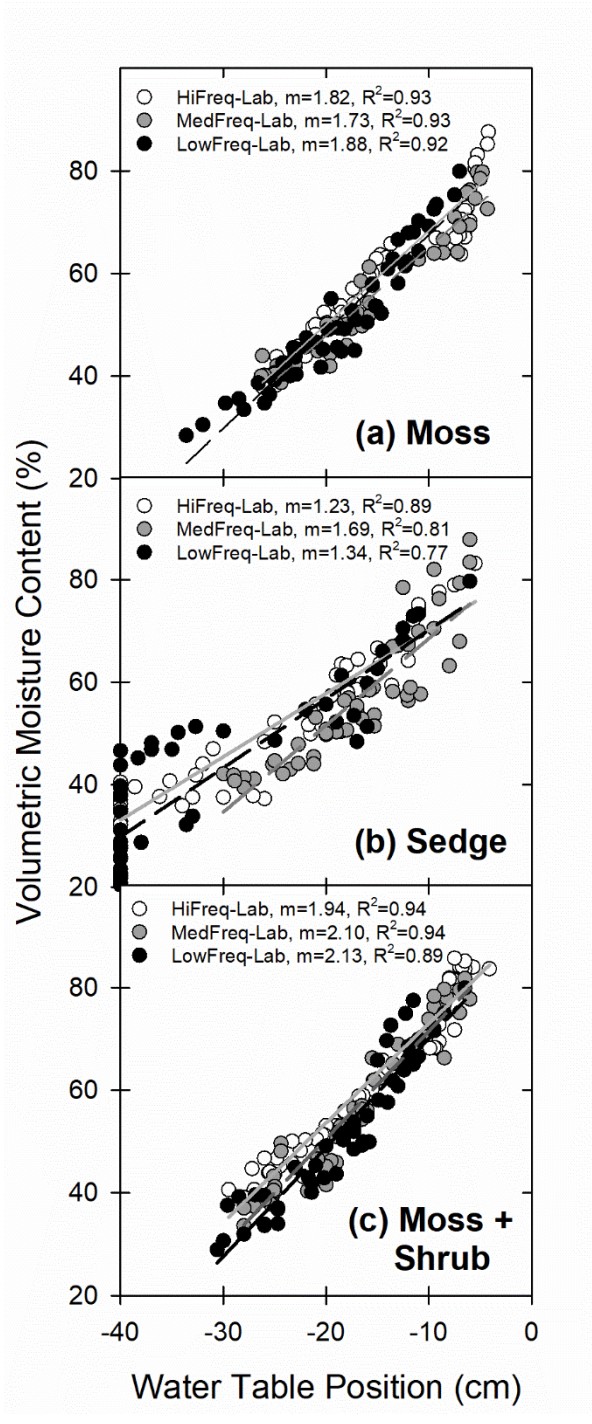

**Figure 2: Relationship between daily average VMC and WT depth for (a) Moss, (b) Sedge + Moss, and (c) Moss + Shrub communities. Slopes and correlation coefficients ($R^2$) of the regressions for each precipitation treatment are shown. All correlations were significant ($p < 0.001$).**



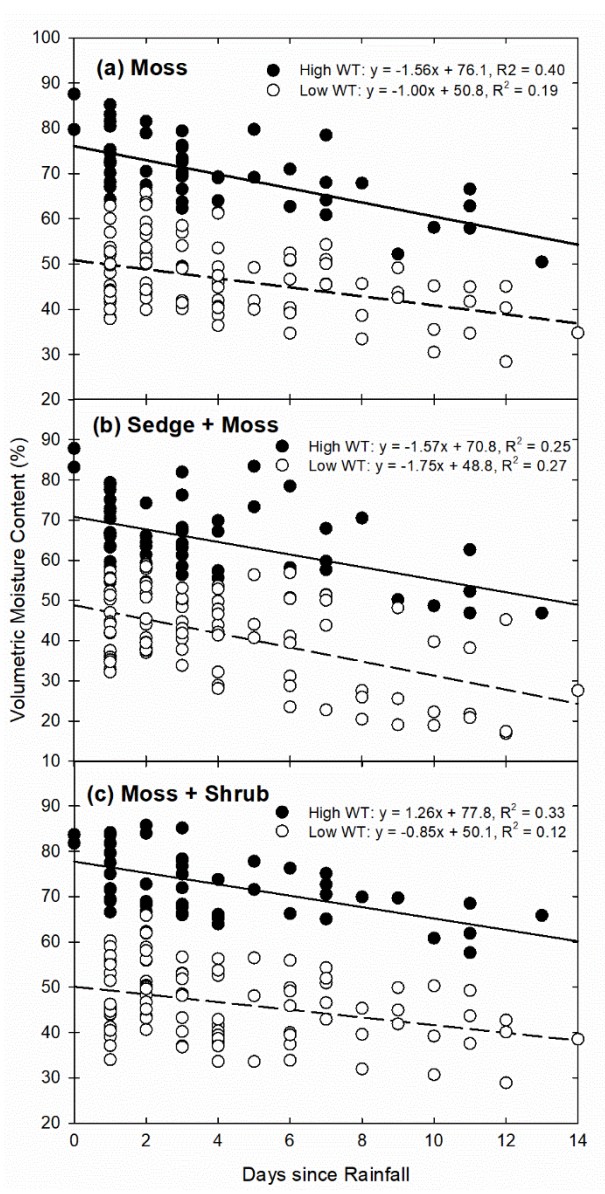

Figure 3: Relationship between daily average VMC and days since the last rainfall-irrigation event for (a) Moss, (b) Sedge + Moss, and (c) Moss + Shrub monoliths. Data for each vegetation type are separated between high-WT and low-WT portions of the experiment. Data are combined between precipitation frequency treatments. Linear regressions are provided. All regressions were significant ($p < 0.05$).





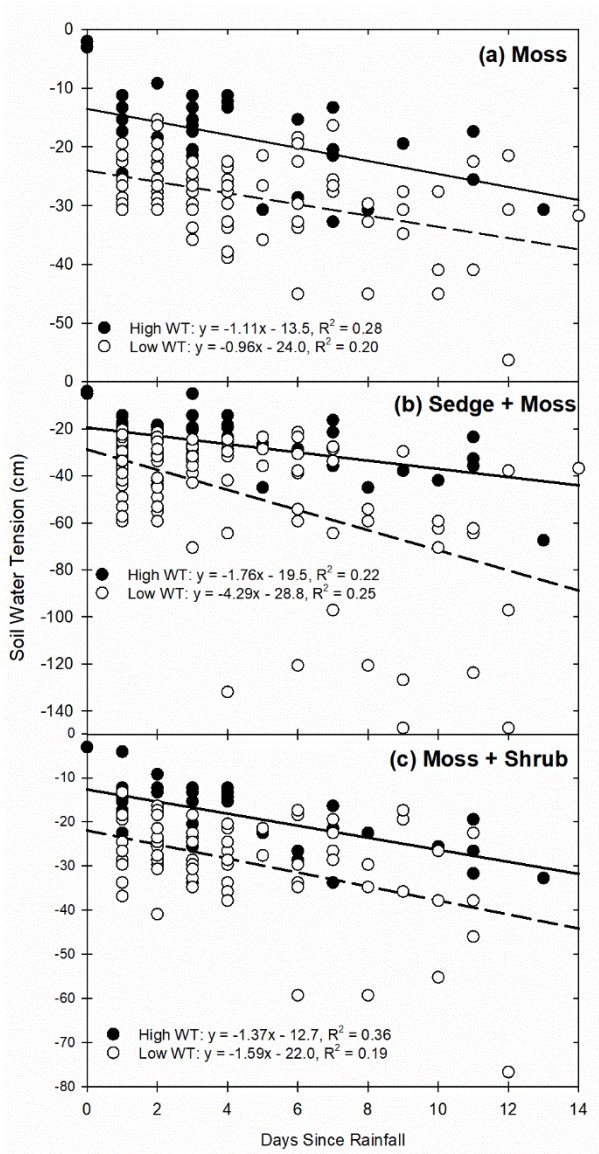

**Figure 4: Relationship between daily soil water tension (cm of water) and days since the last rainfall-irrigation event for (a) Moss, (b) Sedge + Moss, and (c) Moss + Shrub monoliths. Data for each vegetation type are separated between high-WT and low-WT portions of the experiment. Data are combined between precipitation frequency treatments. Linear regressions are provided. All regressions were significant ($p < 0.05$).**





Figure 5: Comparison of mean GEP, ER, and NEE between rainfall treatments within each WT treatment for each vegetation community: a) Moss, b) Moss + Shrub, and c) Sedge + Moss. Error bars represent the standard error of the mean. Negative NEE represents net $CO_2$ uptake. Different letters indicate significant differences ($p < 0.05$) between treatments within each WT period.





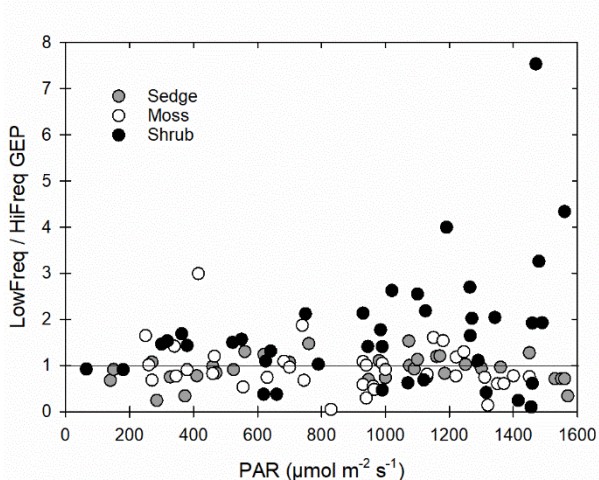

**Figure 6: Ratios of GEP of plots receiving the low-frequency treatment to plots receiving the high-frequency treatment at the same PAR values. Different symbols indicate measurements taken in the Moss, Sedge, and Shrub vegetation communities at the field site during May-September 2015. The horizontal line indicates equal GEP between frequency treatments at the given PAR value.**








**Figure 7: Hydrologic controls on $CO_2$ exchange in *S. capillifolium*-dominated monoliths, depicting relationships in (a) GEP, (b) ER, and (c) NEE between rainfall frequency treatments. Relationships in (a) are unimodal with indicated correlation coefficients and significance. Relationships in (b) are linear and are indicated with correlation coefficients. All regressions in (b) were significant at $p < 0.001$. Relationships for HiFreq-Lab and MedFreq-Lab treatments in (c) are second-order polynomial; relationship for LowFreq-Lab is third-order polynomial with indicated correlation coefficients and significance.**








**Figure 8: Hydrologic controls on CO₂ exchange in *C. oligisperma*-dominated monoliths, depicting relationships in (a) GEP, (b) ER, and (c) NEE between rainfall frequency treatments. Relationship in (a) is unimodal with indicated correlation coefficients and significance. Relationships in (b) and (c) are linear and are indicated with correlation coefficients. All regressions in (b) and (c) were significant at p < 0.001.**






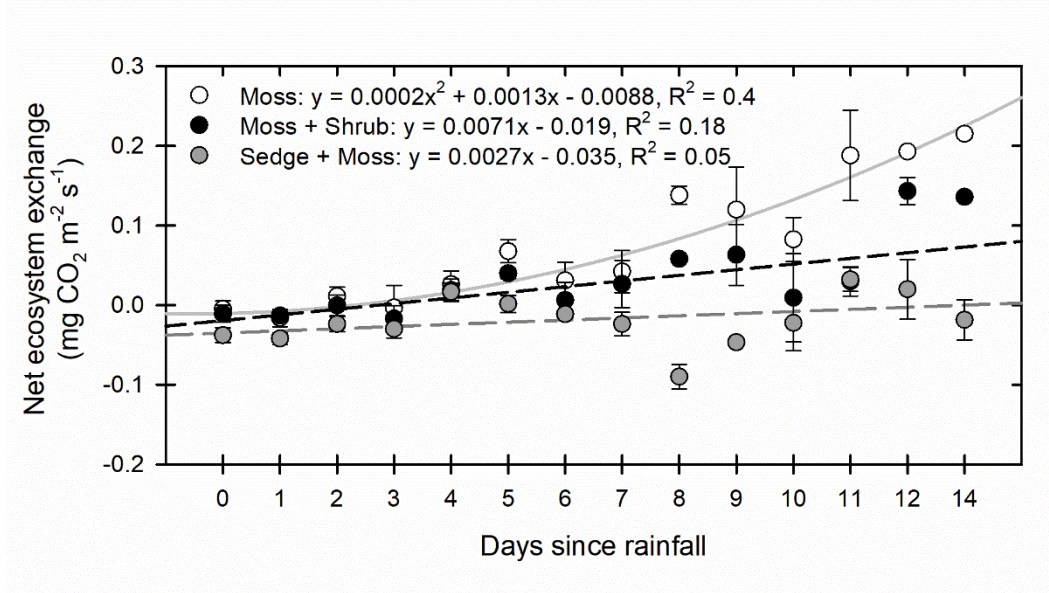


**Figure 9: Relationships between NEE and number of consecutive dry days since rainfall for each vegetation community. Negative NEE represents CO₂ uptake. Error bars represent the standard deviation of the mean. Relationship for the Moss and Moss + Shrub communities are significant at $p < 0.001$; significance for the Sedge + Moss communities is $p = 0.045$.**
