# Peer review of "Response of hydrology and CO₂ flux to experimentally-altered rainfall frequency in a temperate poor fen, southern Ontario, Canada"

_Biogeosciences, 2017_

## Referee Comment (RC1) · Anonymous Referee #1 · 3 Jan 2018

Alongside significant results, the submitted article also provides some important baseline information with concise descriptions which is useful for climate model simulations. The objectives are clear and analyses are also straightforward and easy to comprehend. At the same time, however, I do feel that some parts are missing while reading through the manuscript.

I see the authors describing field experiment which includes the CO2 fluxes measurement (line 105-113) but not discussing the results of this experiment. So, I expect further descriptions about this in the manuscript and also objectives of conducting two

experiments.

The authors mentioned the lowering of WT increases respiration and switching the communities to net CO2 source or C neutral. I think it would be better to elaborate a bit on the mechanisms driving the increased respiration.

I will point out some other minor comments which I feel could be improved.

Line 230: The magnitude of NEE under low WT was greater during frequent rain than during the MedFreq and LowFreq. It would be good to describe the meaning of 'greater NEE' here.

Line 261: It doesn't look like -29 cm in Fig. 8c.

Figure 3c: The negative sign of the slope for high WT is missing.

[Figure]

---

## Referee Comment (RC2) · Anonymous Referee #2 · 17 Jan 2018

This manuscript presents the results of a rainfall frequency manipulation experiment on vegetated peat monoliths' soil moisture characteristics and CO2 fluxes. The authors raise the issue that climate projections predict more intense but less frequent rainfall. In peatlands, particularly those with non-vascular plant species as the main peat "builder", near-surface drying has the potential to greatly influence atmospheric CO2 exchange in these ecosystems. As the authors note, there is a body of literature examining this general topic, however an examination of the combined effects of vegetation community, water table and rainfall frequency on peat moisture characteristics

and CO2 fluxes would be a valuable addition.

There are a number of issues in this manuscript that if addressed, could help this paper make a stronger contribution. The method section includes a description of a field experiment that is mentioned only briefly in the results and discussion. As presented now, there is no need for any mention of the field experiment. However, I wonder how much overlap there is in the background, general research questions and conclusions that are reached in the field experiment manuscript currently in submission (Radu and Duval, submitted). As I note further below, it would seem these are complementary studies that would be much stronger presented together.

Because the peat monoliths are repeatedly measured for a variety of response variables, the data are not independent. I would recommend analysis be carried out using mixed effects models instead of ANOVA.

The authors conclude that less frequent rain and surface drying will give vascular plant species a competitive advantage. There is no data presented in this study that suggests a shift in plant community composition. On what basis are the authors making this conclusion? It would be important to discuss the mechanisms by which competition structures peatland vegetation communities and what might lead to changes in species composition in a fen like the one studied. In addition, I would suggest that the importance of Sphagnum as a peat builder and long-term storage of C is a key issue to raise in addition to shorter-term changes in peatland CO2 exchange.

The variables with negative values (NEE, GPP, WT, soil tension) are not correctly described in a number of instances. For example, more negative NEE means greater net CO2 uptake but the NEE value itself decreased.

The results section largely walks the reader through each figure and table (and there are 2-3 figures that are not necessary). I don't think this level of detail in the text is needed. In the discussion section, the results as presented again but this time in a more readable and informative way. I would recommend starting with the results

statements found in the discussion and add only key details. Currently, the discussion is largely a review of the results with little discussion of mechanism or context within the body of literature on this topic. Similarly, the conclusion summarizes the results again. It might be that presenting the lab and field experiments in separate manuscripts limits what the authors can discuss/conclude, and the manuscript would be much stronger if these complementary studies were presented together.

Specific comments

Line 12: Typically it is volumetric water content or VWC

Line 20: Did your study show that there could be increased vascular plant growth?

Line 38: Should be "season"?

Line 73: And there field studies, e.g. Nijp et al. (2015) Global Change Biology, 21, 2309-2320.

Lines 73-75: Mostly a repeat of previous sentence. Reword.

Line 76: Use acronyms consistently throughout.

Line 93: Remove Section 2.2

Line 134: "integrated" rather than "composite"?

Line 135: Add "(SMS)"

Line 139-144: How soon after was the opaque chamber used? How much did temperature increase in the clear vs. opaque chamber? How might a difference in temperature affect estimates of GPP?

Lines 153-163: Remove.

Line 180: EC5 sensors only accurate to 2-3% VWC. Instead present as per 10 cm drop in WT?

Line 184-197: Given these are monoliths in a lab, are the actual rates important to list? I recommend dropping Figures 2, 3, and 4 and describing trends more briefly.

Line 204: Change "demonstrates" to "illustrates".

Line 234-239: Why analyze ratios?

Line 375: Include the importance of WT position.

Line 610 and 625: Define acronyms and symbols in caption.
* * *

---

## Author Comment (AC1) · 21 Feb 2018

We thank both reviewers for their constructive comments on our submitted manuscript. We have considered each point carefully and made the suggested changes or provided greater detail into our justification for not wholly adopting an idea. Overall, we feel the revised manuscript is more structurally sound and informative for the international biogeoscience community. We have reduced the number of figures by two and removed a table from the main manuscript and moved them to the supplemental material. We have reworked and removed most of the treatment of the field study, as suggested

primarily by Reviewer #2. We have added discussion to the underlying mechanisms driving our results in a number of instances based on comments of Reviewers #1 and #2. Finally, we are in the midst of rerunning our statistical analysis to ensure the correct approach is utilized.

The detailed responses to the reviewers' comments are below, with their comments in quotations and our responses on a separate line preceded by three asterisks: ***. We have also uploaded our working revised manuscript with highlighted changes.

We thank the associate editor, Paul Stoy, for his efforts with handling this manuscript and welcome further feedback.

Thank you,

Tim Duval (on behalf of my co-author, Danielle Radu).

~

"Alongside significant results, the submitted article also provides some important baseline information with concise descriptions which is useful for climate model simulations. The objectives are clear and analyses are also straightforward and easy to comprehend. At the same time, however, I do feel that some parts are missing while reading through the manuscript."

"I see the authors describing field experiment which includes the CO2 fluxes measurement (line 105-113) but not discussing the results of this experiment. So, I expect further descriptions about this in the manuscript and also objectives of conducting two experiments."

***Based on this comment, and some of the comments of Reviewer #2 we have decided to take a different approach than the one suggested by Reviewer #1. We have decided to remove the methodology of the field study, as well as the presentation of the relevant results of the field study in the present manuscript (Figure 6, Results L234-239). The field study is an article currently in press; in the article we present seasonal changes

to vegetation and model CO2 balances as a consequence of altered rainfall pattern. The field data presented in this study were analysed the same way as the lab study, and was meant to support the controlled-environment of the lab study. There was no separate objective to the field study.

"The authors mentioned the lowering of WT increases respiration and switching the communities to net CO2 source or C neutral. I think it would be better to elaborate a bit on the mechanisms driving the increased respiration."

***We have added text to the discussion (L323-326 and L329-331) to highlight the mechanisms of increased respiration with lowered WT due to greater aeration.

"I will point out some other minor comments which I feel could be improved."

"Line 230: The magnitude of NEE under low WT was greater during frequent rain than during the MedFreq and LowFreq. It would be good to describe the meaning of 'greater NEE' here."

***We have added a clause to this sentence to highlight that the greater magnitude of NEE for the HiFreq treatment refers to more net carbon uptake (new Line 243).

"Line 261: It doesn't look like -29 cm in Fig. 8c."

***The lowest NEE rate (least negative, -29 cm) stated for the HiFreq treatment in Fig. 8c is the computed value of the quadratic regression model. We agree that the data cloud doesn't necessarily agree with this computed value. We also note that while significant ($p < 0.001$) the correlation was rather weak ($R^2 = 0.15$). Thus, we have modified this statement to depict the more general trend of the data.

"Figure 3c: The negative sign of the slope for high WT is missing."

***Thank you for noticing this omission. The negative sign will be added to our final submission.

Please also note the supplement to this comment:
https://www.biogeosciences-discuss.net/bg-2017-485/bg-2017-485-AC1-supplement.pdf

―――――――――――――

---

## Author Comment (AC2) · 21 Feb 2018

We thank both reviewers for their constructive comments on our submitted manuscript. We have considered each point carefully and made the suggested changes or provided greater detail into our justification for not wholly adopting an idea. Overall, we feel the revised manuscript is more structurally sound and informative for the international biogeoscience community. We have reduced the number of figures by two and removed a table from the main manuscript and moved them to the supplemental material. We have reworked and removed most of the treatment of the field study, as suggested

primarily by Reviewer #2. We have added discussion to the underlying mechanisms driving our results in a number of instances based on comments of Reviewers #1 and #2. Finally, we are in the midst of rerunning our statistical analysis to ensure the correct approach is utilized.

The detailed responses to the reviewers' comments are below, with their comments in quotations and our responses on a separate line preceded by three asterisks: ***. We have also uploaded our working revised manuscript with highlighted changes.

We thank the associate editor, Paul Stoy, for his efforts with handling this manuscript and welcome further feedback.

Thank you,

Tim Duval (on behalf of my co-author, Danielle Radu).

~~

"This manuscript presents the results of a rainfall frequency manipulation experiment on vegetated peat monoliths' soil moisture characteristics and CO2 fluxes. The authors raise the issue that climate projections predict more intense but less frequent rainfall. In peatlands, particularly those with non-vascular plant species as the main peat "builder", near-surface drying has the potential to greatly influence atmospheric CO2 exchange in these ecosystems. As the authors note, there is a body of literature examining this general topic, however an examination of the combined effects of vegetation community, water table and rainfall frequency on peat moisture characteristics and CO2 fluxes would be a valuable addition."

"There are a number of issues in this manuscript that if addressed, could help this paper make a stronger contribution. The method section includes a description of a field experiment that is mentioned only briefly in the results and discussion. As presented now, there is no need for any mention of the field experiment. However, I wonder how much overlap there is in the background, general research questions and conclusions

that are reached in the field experiment manuscript currently in submission (Radu and Duval, submitted). As I note further below, it would seem these are complementary studies that would be much stronger presented together."

***As the reviewer suggests, the present manuscript, and our field study currently in press are directly related; however, we feel there is significant departure between the two, which warranted two manuscripts. Our article in press documents the seasonal change in species composition in the field and models the seasonal response of altered rainfall regime on GEP, ER, and NPP. The field study did include WT and VMC changes in the seasonal $CO_2$ models, but we did not directly explore the effect of rain frequency on hydrology (& couldn't adequately measure field-level ET and soil tension). The present manuscript submission provides the process-based interactions between hydrology and $CO_2$ dynamics as affected by the changing rainfall regime that elicited the observed vegetation and seasonal $CO_2$ balance changes. Our intent with including the field data here was to strengthen our argument based on the lab monolith results, that what was observed in the controlled lab setting also happened in the field. Based on the reviewer's comments throughout we feel this approach may unintentionally obfuscate our overall study goals. We agree to the removal of almost all mention of the field study. We now present the article in press as setting the stage for the current manuscript, but we do comment briefly on the field data presented here (whilst moving all field-related tables and figures to the Supplemental Material).

"Because the peat monoliths are repeatedly measured for a variety of response variables, the data are not independent. I would recommend analysis be carried out using mixed effects models instead of ANOVA."

***We will perform this method of statistical testing for the final submission (I'm still trying to make sure I haven't made any mistakes with the formatting of the data file).

"The authors conclude that less frequent rain and surface drying will give vascular plant species a competitive advantage. There is no data presented in this study that suggests a shift in plant community composition. On what basis are the authors making this conclusion? It would be important to discuss the mechanisms by which competition structures peatland vegetation communities and what might lead to changes in species composition in a fen like the one studied. In addition, I would suggest that the importance of Sphagnum as a peat builder and long-term storage of C is a key issue to raise in addition to shorter-term changes in peatland $CO_2$ exchange."

***While we do not document community shifts in the present manuscript, we do document increased stress on Sphagnum in the presence of sedges through increased tension in the near-surface zone, in addition to differential effects on GEP between plant functional types. To avoid confusion with the concept of community ecology competition we have toned-down this language. We thank the reviewer for the suggestion to incorporate the effects on long-term storage of carbon. We have added a paragraph to our final discussion section (L382-391).

"The variables with negative values (NEE, GPP, WT, soil tension) are not correctly described in a number of instances. For example, more negative NEE means greater net $CO_2$ uptake but the NEE value itself decreased."

***We have reread the manuscript and rewritten a few instances of phrasing where we agree the negative values were interpreted incorrectly. We will do so again before final submission to ensure we have not missed any.

"The results section largely walks the reader through each figure and table (and there are 2-3 figures that are not necessary). I don't think this level of detail in the text is needed. In the discussion section, the results as presented again but this time in a more readable and informative way. I would recommend starting with the results statements found in the discussion and add only key details. Currently, the discussion is largely a review of the results with little discussion of mechanism or context within the body of literature on this topic. Similarly, the conclusion summarizes the results again. It might be that presenting the lab and field experiments in separate manuscripts limits

what the authors can discuss/conclude, and the manuscript would be much stronger if these complementary studies were presented together."

***We thank the reviewer for their views with respect to the style of information presentation. We acknowledge that results are increasingly being quickly summarized in the results sections of many articles, with authors presenting only the general trends and/or key findings of their datasets. We do not follow this approach, as we feel it limits the usefulness of the data to only specifically what the author(s) wish the readers to pick up from the information. We point out Reviewer #1's comment: "the submitted article also provides some important baseline information with concise descriptions which is useful for climate model simulations." We did not write this manuscript with the intent of providing specific data to feed into climate model simulations, but in our detailed accounting of our data we have provided information that others may find useful in their advancement of biogeosciences. We respectfully wish to leave the level of detail in the results section as currently constructed.

We do not agree with the reviewer's statement that we did not contextualize our results within the body of literature on this topic. In our discussion we originally had related the results of our study to 33 individual articles from the associated literature. Both reviewers have pointed out areas in the discussion that would benefit from greater mechanistic explanations, and we have provided these, and additional references.

"Specific comments"

"Line 12: Typically it is volumetric water content or VWC"

***We feel this distinction is perhaps one of geographic or institutional preference. We use the term "moisture" rather than "water" in line with the concept of relating VMC to soil tension with the soil moisture characteristic curve. We have scanned several peatland articles dealing with soil moisture and found a relatively equal split between the use of VMC and VWC. We hope it is okay to keep our preferred term VMC.

"Line 20: Did your study show that there could be increased vascular plant growth?"

***We have corrected this line to speak only about the observed increase in sedge GEP

"Line 38: Should be "season"?"

***Thank you for noticing this mistake, we have deleted the "s".

"Line 73: And there field studies, e.g. Nijp et al. (2015) Global Change Biology, 21, 2309-2320."

***We have added this important reference.

"Lines 73-75: Mostly a repeat of previous sentence. Reword."

***We have modified parts of both sentences to avoid the repetition. We have also added two sentences between the two in question to highlight our previous, field-based, work.

"Line 76: Use acronyms consistently throughout."

***We have made the required changes.

"Line 93: Remove Section 2.2"

***We have removed this section.

"Line 134: "integrated" rather than "composite"?"

***We have made this change.

"Line 135: Add "(SMS)""

***We incorrectly stated the Infield 7 tensicorder was made by Soil Measurement Systems, which we think was the reason for the request to put the abbreviation in parentheses after the company's first mention. We have correctly identified the manufacturer as UMS, which we hope nullifies the need for this request.

"Line 139-144: How soon after was the opaque chamber used? How much did temperature increase in the clear vs. opaque chamber? How might a difference in temperature affect estimates of GPP?"

***We have added some explanatory text on this matter (L152-155). All the monoliths were measured for NEE first, then the opaque chamber was used for ER on all the monoliths. The time between a monolith being measured for NEE and then for ER fluctuated between 20 and 60 minutes. As indicated in the new text, the temperature only increased marginally during a run (1.5 degrees, at most). We are confident our methods did not induce great increases in either carbon uptake or respiration, particularly in the controlled environmental conditions.

"Lines 153-163: Remove."

***We have removed this section

"Line 180: EC5 sensors only accurate to 2-3% VWC. Instead present as per 10 cm drop in WT?"

***We thank you for this suggestion, and have adopted this presentation.

"Line 184-197: Given these are monoliths in a lab, are the actual rates important to list? I recommend dropping Figures 2, 3, and 4 and describing trends more briefly."

***We are not sure we fully understand this comment. It is unclear to us why rates found in the lab are not important to list. We feel the differing rates of moisture and tension decreases with greater consecutive dry days between the high and low WT treatments, and in particular the differences between the vegetation types are relevant to the study objectives. We have chosen to leave Figures 3 and 4 in the manuscript. We do see the case of removing Figure 2 from the story. We have moved Figure 2 to the supplemental material, and described the trends contained therein only generally.

"Line 204: Change "demonstrates" to "illustrates"."

***This change has been made

"Line 234-239: Why analyze ratios?"

***Our original intent was to comment on the differential effect of high light in the field on GEP between high- and low-frequency rain treatments, and found the ratio between the two treatments a good approach. However, we acknowledge that this excludes the medium-frequency treatment, and more importantly, pertains exclusively to the field data, which is now mostly removed from the study. Also, we do acknowledge this portion of the manuscript is somewhat unrelated to the main study objectives. We have removed this section of the text and moved Figure 6 to the Supplemental Material.

"Line 375: Include the importance of WT position."

***We agree that the WT position has a strong impact on the $CO_2$ dynamics in our study. We chose to highlight this importance separately in the final line of the first paragraph of the conclusion (L399-401).

"Line 610 and 625: Define acronyms and symbols in caption."

***We have added the acronyms and symbols to the caption.

Please also note the supplement to this comment:
https://www.biogeosciences-discuss.net/bg-2017-485/bg-2017-485-AC2-supplement.pdf

**Supplement:**

[revised manuscript text omitted]

**Commented [TD1]:** We will redo the stats using the mixed effects modelling approach suggested by reviewer #2.

[revised manuscript text omitted]

---

## Author Response (AR2)

Response to Reviewer #3's comments for the revision of:

BG-2017-485:

5   Response of hydrology and CO2 flux to experimentally-altered rainfall frequency in a temperate poor fen, southern Ontario, Canada

Dear Associate Editor Paul Stoy,

10   Please accept this document along with the relevant uploads as our response to reviewer #3's comments for the above referenced manuscript.

The detailed responses to the reviewer's comments are below, with their comments in bold and our responses in normal type. Our revised manuscript with highlighted changes follows our responses.

Once again, thank you for your efforts in handling this manuscript and hope you are pleased with our revisions suggested by all three of the reviewers.

Thank you,

Tim Duval (on behalf of my co-author, Danielle Radu).

**Reviewer #3:**

**My biggest question is, have peatland areas experienced changes in precipitation regime? This popular press article would suggest that they have over parts of their distribution: https://fivethirtyeight.com/features/the-midwest-is-getting-drenched-and-its-causing-big-problems/**

**Any demonstration of changing precipitation regimes in peatlands would help, perhaps as a small analysis in the results section to discuss consequences for the site in the discussion.**

There is increasing evidence for changes to precipitation regime globally, including the main peatland-containing regions; however, the form of most of the analysis is not directly compatible with the goals of our manuscript. The results of other studies, including the many that we cite in the introduction and discussion, are concerned with either changes in precipitation totals or changes in extreme precipitation events, which are universally depicted as high magnitude events. The research is fairly conclusive that the frequency of extreme events is increasing. What has not been directly addressed (as far as we can tell) is whether the length of time between these extreme events is also increasing. Our preliminary estimation suggests this is occurring: total rain has increased by ~10 % but high-magnitude events have increased at a greater rate, resulting in more days without rain. Certainly the general consensus of climate models is the prediction of more extreme rain events interspersed with longer dry periods. We thank the reviewer for this comment; however, we feel devoting a section of this manuscript to global/regional precipitation trends would detract from the main focus of our paper. We do agree this is an important issue, and are in the midst of doing just such an analysis (for our study region at least, and hopefully for surrounding Great Lake states and provinces). We do point out though, that the popular press article provided focuses on extreme events, with little mention of the timing / distribution of these events. Our paper makes the point that it is primarily the intervening dry periods associated with this regime change that is most crucial for peatland ecosystems, as the resultant near-surface moisture stress is detrimental to the moss communities. We have addressed this important point of the reviewer with a rewording and addition to a paragraph in our discussion (L318-327 (L412-421 in this document)), providing more evidence of a change in the precipitation regime in our study region and pointing out the lack of research on periods without rain during the growing season.

**Minor points follow:**

**'Repackaging' is colloquial on page 2 line 37.**

We have replaced this word with "redistribution".

**On p 3 L 75 it would help the reader to succinctly describe the findings of Radu and Duval (2018) so the reader can distinguish between it in the present paper.**

We have rewritten this sentence to provide the two main findings from our previous paper.

**Why 'rainwater solution additions' on page 4?**

65

We are not entirely sure we understand the question. Our interpretation is that the reviewer is confused as to what our term is referencing. We were stating that we measured the mass of the monoliths both before and after we delivered our prescribed watering treatments. We have slightly reworded this sentence to delete the term "solution" (which we used to reference that there were dilute minerals and nutrients
70 added as described above) to increase clarity (we hope).

**The abbreviation 'CDD' is introduced for the first time in the last paragraph. Don't bother introducing this new term at this point in the manuscript.**

75 We have removed the abbreviation and changed an instance of the occurrence of the term "consecutive dry days".

**Note subscript for $CO_2$ in the legend of table 2. A brief read for minor details like this would benefit the manuscript.**
80

We thank the reviewer for spotting this error. We have corrected this and reread the manuscript and corrected what we hope are all the remaining typographical errors.

**Figure 1 would benefit from multiple subplots (three, for moss, shrub, and sedge) and perhaps the**
85 **judicious use of color. At the moment it is hard to read and why is 60% highlighted? Is this the value of VMC at saturation?**

We have broken this figure into the three panels, as suggested. We agree that this greatly increases the readability of this figure. The VMC of 60 % was originally highlighted because that was the VMC at
90 which the Moss monoliths under the high frequency treatment switched from net carbon uptake to net carbon loss. We realize that this was only one of nine vegetation – rainfall treatment interactions, and that for completeness each treatment should have its own guiding line. However, even with the now three panels the three vertical guide lines distract the reader from the message of the figure. As such, we have removed the vertical 60 % VMC line from the revised figure.

[revised manuscript text omitted]